# DIFFUSION SCATTERING TRANSFORMS ON GRAPHS

**Fernando Gama**
Dept. of Electrical and Systems Engineering
University of Pennsylvania

**Alejandro Ribeiro**
Dept. of Electrical and Systems Engineering
University of Pennsylvania

**Joan Bruna**
Courant Institute of Mathematical Sciences
New York University

## ABSTRACT

Stability is a key aspect of data analysis. In many applications, the natural notion of stability is geometric, as illustrated for example in computer vision. Scattering transforms construct deep convolutional representations which are certified stable to input deformations. This stability to deformations can be interpreted as stability with respect to changes in the metric structure of the domain.

In this work, we show that scattering transforms can be generalized to non-Euclidean domains using diffusion wavelets, while preserving a notion of stability with respect to metric changes in the domain, measured with diffusion maps. The resulting representation is stable to metric perturbations of the domain while being able to capture "high-frequency" information, akin to the Euclidean Scattering.

## 1 INTRODUCTION

Convolutional Neural Networks (CNN) are layered information processing architectures. Each of the layers in a CNN is itself the composition of a convolution operation with a pointwise nonlinearity where the filters used at different layers are the outcome of a data-driven optimization process (LeCun et al., 2010; 2015). Scattering transforms have an analogous layered architecture but differ from CNNs in that the convolutional filters used at different layers are not trained but selected from a multi-resolution filter bank (Mallat, 2012; Bruna & Mallat, 2013). The fact that they are not trained endows scattering transforms with intrinsic value in situations where training is impossible – and inherent limitations in the converse case. That said, an equally important value of scattering transforms is that by isolating the convolutional layered architecture from training effects it permits analysis of the fundamental properties of CNN information processing architectures. This analysis is undertaken in Mallat (2012); Bruna & Mallat (2013) where the fundamental conclusion is about the stability of scattering transforms with respect to deformations in the underlying domain that are close to translations.

In this paper we consider graphs and signals supported on graphs such as brain connectivity networks and functional activity levels (Huang et al., 2016), social networks and opinions (Jackson, 2008), or user similarity networks and ratings in recommendation systems (Huang et al., 2018). Our specific goals are: (i) To define a family of graph-scattering transforms. (ii) To define a notion of deformation for graph signals. (iii) To study the stability of graph scattering transforms with respect to this notion of deformation. To accomplish goal (i) we consider the family of graph diffusion wavelets which provide an appropriate construction of a multi-resolution filter bank (Coifman & Maggioni, 2006). Our diffusion scattering transforms are defined as the layered composition of diffusion wavelet filter banks and pointwise nonlinearities. To accomplish goal (ii) we adopt the graph diffusion distance as a measure of deformation of the underlying domain (Coifman & Lafon, 2006; Nadler et al., 2006). Diffusion distances measure the similarity of two graphs through the time it takes for a signal to be diffused on the graph. The major accomplishment of this paper is to show that the diffusion graph scattering transforms are stable with respect to deformations as measured with respect to diffusion distances. Specifically, consider a signal $\mathbf{x}$ supported on graph $G$ whose diffusion scattering transform is denoted by the operator $\mathbf{\Psi}_G$. Consider now a deformation of the

signal's domain so that the signal's support is now described by the graph $G'$ whose diffusion scattering operator is $\mathbf{\Psi}_{G'}$. We show that the operator norm distance $\|\mathbf{\Psi}_G - \mathbf{\Psi}_{G'}\|$ is bounded by a constant multiplied by the diffusion distance between the graphs $G$ and $G'$. The constant in this bound depends on the spectral gap of $G$ but, very importantly, does not depend on the number of nodes in the graph.

It is important to point out that finding stable representations is not difficult. E.g., taking signal averages is a representation that is stable to domain deformations – indeed, invariant. The challenge is finding a representation that is stable and rich in its description of the signal. In our numerical analyses we show that linear filters can provide representations that are either stable or rich but that cannot be stable and rich at the same time. The situation is analogous to (Euclidean) scattering transforms and is also associated with high frequency components. We can obtain a stable representation by eliminating high frequency components but the representation loses important signal features. Alternatively, we can retain high frequency components to have a rich representation but that representation is unstable to deformations. Diffusion scattering transforms are observed to be not only stable – as predicted by our theoretical analysis – but also sufficiently rich to achieve good performance in graph signal classification examples.

## 2 RELATED WORK

Since graph and graph signals are of increasing interest but do not have the regular structure that would make use of CNNs appealing, it is pertinent to ask the question of what should be an appropriate generalization of CNNs to graphs and the graph signals whose topology they describe (Bronstein et al., 2017). If one accepts the value of convolutions as prima facie, a natural solution is to replace convolutions with graph shift invariant filters which are known to be valid generalizations of (convolutional) time invariant filters (Bruna et al., 2014). This idea is not only natural but has been demonstrated to work well in practical implementations of Graph Neural Networks (GNNs) (Defferrard et al., 2016; Gama et al., 2019; Gilmer et al., 2017; Henaff et al., 2015; Kipf & Welling, 2017). Same as Euclidean scattering transforms, our graph scattering transforms differ from GNNs in that they do not have to be trained. The advantages and limitations of the absence of training notwithstanding, our work also sheds light on the question of why graph convolutions are appropriate generalizations of regular domain convolutions for signal classification problems. Our work suggests that the value of GNNs stems from their stability relative to deformations of the underlying domain that are close to permutations – which is the property that a pair of graphs must satisfy to have small diffusion distance.

The stability results obtained in this paper build on the notion of scattering transforms. These scattering representations were introduced by Mallat (2012) and further developed in Bruna & Mallat (2013) with computer vision applications. Since, these representations have been extended to handle transformations on more complex groups, such as roto-translations (Sifre & Mallat, 2013; Oyallon & Mallat, 2015), and to domains such as audio processing (Andén & Mallat, 2014) and quantum chemistry (Eickenberg et al., 2017).

Similarly as in this work, extensions of scattering to general graphs have been considered in Chen et al. (2014) and Zou & Lerman (2018). Chen et al. (2014) focuses on Haar wavelets that hierarchically coarsen the graph, and relies on building multiresolution pairings. The recent Zou & Lerman (2018) is closest to our work. There, the authors define graph scattering using spectrally constructed wavelets from (Hammond et al., 2011), and establish some properties of the resulting representation, such as energy conservation and stability to spectral perturbations. In contrast, our stability results are established with respect to diffusion metric perturbations, which are generally weaker, in the sense that they define a weaker topology (see Section 3). We use diffusion wavelets (Coifman & Maggioni, 2006) to obtain multi-resolution graph filter banks that are localized in frequency as well as in the graph domain, while spanning the whole spectrum. Diffusion wavelets serve as the constructive basis for the obtained stability results. Our work is also closely related to recent analysis of stability of Graph Neural Networks in the context of surface representations in (Kostrikov et al., 2017). In our work, however, we do not rely on extrinsic deformations and exploit the specific multiresolution structure of wavelets.

## 3 PROBLEM SET-UP

This section introduces our framework and states the desired stability properties of signal representations defined on general non-Euclidean domains.

### 3.1 EUCLIDEAN STABILITY TO DEFORMATIONS WITH SCATTERING

Motivated by computer vision applications, our analysis starts with the notion of deformation stability. If $\mathbf{x}(u) \in L^2(\Omega)$ is an image defined over an Euclidean domain $\Omega \subset \mathbb{R}^d$, we are interested in signal representations $\Phi : L^2(\Omega) \to \mathbb{R}^K$ that are stable with respect to small deformations. If $\mathbf{x}_\tau(u) := \mathbf{x}(u - \tau(u))$ denotes a change of variables with a differentiable field $\tau : \Omega \to \Omega$ such that $\|\nabla \tau\| < 1$, then we ask

$$\forall \, \mathbf{x}, \tau \, , \, \|\Phi(\mathbf{x}) - \Phi(\mathbf{x}_\tau)\| \lesssim \|\mathbf{x}\| \|\tau\| \, , \text{with} \tag{1}$$

$\|\tau\| := \|\nabla \tau\|_\infty$ denoting a uniform bound on the operator norm of $\nabla \tau$. In this setting, a notorious challenge to achieving (1) while keeping enough discriminative power in $\Phi(\mathbf{x})$ is to transform the high-frequency content of $\mathbf{x}$ in such a way that it becomes stable.

Scattering transforms (Mallat, 2012; Bruna & Mallat, 2013) provide such representations by cascading wavelet decompositions with pointwise modulus activation functions. We briefly summarize here their basic definition. Given a mother wavelet $\psi \in L^1(\Omega)$ with at least a vanishing moment $\int \psi(u)du = 0$ and with good spatial localization, we consider rotated and dilated versions $\psi_{j,c}(u) = 2^{-jd}\psi(2^{-j}R_c u)$ using scale parameter $j$ and angle $\theta \in \{2\pi c/C\}_{c=0,\dots,C-1}$. A wavelet decomposition operator is defined as a filter bank spanning all scales up to a cutoff $2^J$ and all angles: $\boldsymbol{\Psi}_J : \mathbf{x} \mapsto (\mathbf{x} * \psi_{j,c})_{j \leq J, c \leq C}$. This filter bank is combined with a pointwise modulus activation function $\rho(z) = |z|$, as well as a low-pass average pooling operator $U$ computing the average over the domain. The resulting representation using $m$ layers becomes

$$\begin{aligned} \Phi(\mathbf{x}) &= \{S_0(\mathbf{x}), S_1(\mathbf{x}), \dots, S_{m-1}(\mathbf{x})\} \, , \text{with} \\ S_k(\mathbf{x}) &= U\rho\boldsymbol{\Psi}_J\rho\dots\boldsymbol{\Psi}_J\mathbf{x} = \{U(||\mathbf{x} * \psi_{\alpha_1}| * \psi_{\alpha_2}| \cdots * \psi_{\alpha_k}|);\}_{\alpha_1,\dots,\alpha_k} \quad (k = 0, \dots, m-1). \end{aligned} \tag{2}$$

The resulting signal representation has the structure of a CNN, in which feature maps are not recombined with each other, and trainable filters are replaced by multiscale, oriented wavelets. It is shown in Mallat (2012) that for appropriate signal classes and wavelet families, the resulting scattering transform satisfies a deformation stablity condition of the form (1), which has been subsequently generalised to broader multiresolution families (Wiatowski & Bölcskei, 2018). In essence, the mechanism that provides stability is to capture high-frequency information with the appropriate spatio-temporal tradeoffs, using spatially localized wavelets.

### 3.2 DEFORMATIONS AND METRIC STABILITY

Whereas deformations provide the natural framework to describe geometric stability in Euclidean domains, their generalization to non-Euclidean, non-smooth domains is not straightforward. Let $\mathbf{x} \in L^2(\mathcal{X})$. If $\mathcal{X}$ is embedded into a low-dimension Euclidean space $\Omega \subset \mathbb{R}^d$, such as a 2-surface within a three-dimensional space, then one can still define meaningful deformations on $\mathcal{X}$ via *extrinsic* deformations of $\Omega$ (Kostrikov et al., 2017).

However, in this work we are interested in intrinsic notions of geometric stability, that do not necessarily rely on a pre-existent low-dimensional embedding of the domain. The change of variables $\varphi(u) = u - \tau(u)$ defining the deformation can be seen as a perturbation of the Euclidean metric in $L^2(\mathbb{R}^d)$. Indeed,

$$\langle \mathbf{x}_\tau, \mathbf{y}_\tau \rangle_{L^2(\mathbb{R}^d, \mu)} = \int_{\mathbb{R}^d} \mathbf{x}_\tau(u) \mathbf{y}_\tau(u) d\mu(u) = \int_{\mathbb{R}^d} \mathbf{x}(u) \mathbf{y}(u) |I - \nabla \tau(u)| d\mu(u) = \langle \mathbf{x}, \mathbf{y} \rangle_{L^2(\mathbb{R}^d, \tilde{\mu})} \, ,$$

with $d\tilde{\mu}(u) = |I - \nabla \tau(u)| d\mu(u)$, and $|I - \nabla \tau(u)| \approx 1$ if $\|\nabla \tau\|$ is small, where $I$ is the identity. Therefore, a possible way to extend the notion of deformation stability to general domains $L^2(\mathcal{X})$ is to think of $\mathcal{X}$ as a metric space and reason in terms of stability of $\Phi : L^2(\mathcal{X}) \to \mathbb{R}^K$ to *metric changes* in $\mathcal{X}$. This requires a representation that can be defined on generic metric spaces, as well as a criteria to compare how close two metric spaces are.

### 3.3 Diffusion Wavelets and Metrics on Graphs

Graphs are flexible data structures that enable general metric structures and modeling non-Euclidean domains. The main ingredients of the scattering transform can be generalized using tools from computational harmonic analysis on graphs. We note that, unlike the case of Euclidean domains, where deformations are equivalent whether they are analyzed from the function domain or its image, in the case of graphs, we focus on deformations on the underlying graph domain, while keeping the same function mapping (i.e. we model deformations as a change of the underlying graph support and analyze how this affects the interaction between the function mapping and the graph).

In particular, diffusion wavelets (Coifman & Maggioni, 2006) provide a simple framework to define a multi-resolution analysis from powers of a diffusion operator defined on a graph. A weighted, undirected graph $G = (V, E, W)$ with $|V| = n$ nodes, edge set $E$ and adjacency matrix $W \in \mathbb{R}^{n \times n}$ defines a diffusion process $A$ in its nodes, given in its symmetric form by the normalized adjacency

$$A := D^{-1/2} W D^{-1/2} \,, \text{ with } D = \operatorname{diag}(d_1, \ldots, d_n) \,, \tag{3}$$

where $d_i = \sum_{(i,j) \in E} W_{i,j}$ denotes the degree of node $i$. Denote by $\mathbf{d} = W\mathbf{1}$ the degree vector containing $d_i$ in the $i$th element. By construction, $A$ is well-localized in space (it is nonzero only where there is an edge connecting nodes), it is self-adjoint and satisfies $\|A\| \leq 1$, where $\|A\|$ is the operator norm. Let $\lambda_0 \geq \lambda_1 \geq \ldots \lambda_{n-1}$ denote its eigenvalues in decreasing order. Defining $\mathbf{d}^{1/2} = (\sqrt{d_1}, \ldots, \sqrt{d_n})$, one can easily verify that the normalized squared root degree vector $\mathbf{v} = \mathbf{d}^{1/2}/\|\mathbf{d}^{1/2}\|_2 = \mathbf{d}/\|\mathbf{d}\|_1$ is the eigenvector with associated eigenvalue $\lambda_0 = 1$. Also, note that $\lambda_{n-1} = -1$ if and only if $G$ has a connected component that is non-trivial and bipartite (Chung, 1997).

In the following, it will be convenient to assume that the spectrum of $A$ (which is real and discrete since $A$ is self-adjoint and in finite-dimensions) is non-negative. Since we shall be taking powers of $A$, this will avoid folding negative eigenvalues into positive ones. For that purpose, we adopt the so-called *lazy diffusion*, given by $T = \frac{1}{2}(I + A)$. In Section 4 we use this diffusion operator to define both a multiscale wavelet filter bank and a low-pass average pooling, leading to the diffusion scattering representation.

This diffusion operator can also be used to construct a metric on $G$. The so-called *diffusion distances* (Coifman & Lafon, 2006; Nadler et al., 2006) measure distances between two nodes $x, x' \in V$ in terms of their associated diffusion at time $s$: $d_{G,s}(x, x') = \|T_G^s \delta_x - T_G^s \delta_{x'}\|$, where $\delta_x$ is a vector with all zeros except a 1 in position $x$.

In this work, we build on this diffusion metric to define a distance between two graphs $G, G'$. Assuming first that $G$ and $G'$ have the same size, the simplest formulation is to compare the diffusion metric generated by $G$ and $G'$ up to a node permutation:

**Definition 3.1.** *Let $G = (V, E, W)$, $G' = (V', E', W')$ have the same size $|V| = |V'| = n$. The normalized diffusion distance between graphs $G$, $G'$ at time $s > 0$ is*

$$\mathrm{d}^s(G, G') := \inf_{\Pi \in \Pi_n} \|(T_G^s)^*(T_G^s) - \Pi^\mathsf{T}(T_{G'}^s)^*(T_{G'}^s)\Pi\| = \inf_{\Pi \in \Pi_n} \|T_G^{2s} - \Pi^\mathsf{T} T_{G'}^{2s}\Pi\| \,, \tag{4}$$

*where $\Pi_n$ is the space of $n \times n$ permutation matrices.*

The diffusion distance is defined at a specific time $s$. As $s$ increases, this distance becomes weaker[1], since it compares points at later stages of diffusion. The role of time is thus to select the smoothness of the 'graph deformation', similarly as $\|\nabla \tau\|$ measures the smoothness of the deformation in the Euclidean case. For convenience, we denote $\mathrm{d}(G, G') = \mathrm{d}^{1/2}(G, G')$ and use the distance at $s = 1/2$ as our main deformation measure. The quantity $\mathrm{d}$ defines a distance between graphs (seen as metric spaces) yielding a stronger topology than other alternatives such as the Gromov-Hausdorff distance, defined as

$$d_{\mathrm{GH}}^s(G, G') = \inf_{\Pi} \sup_{x, x' \in V} |d_G^s(x, x') - d_{G'}^s(\pi(x), \pi(x'))|$$

with $d_G^s(x, x') = \|T_G^t(\boldsymbol{\delta}_x - \boldsymbol{\delta}_{x'})\|_{L^2(G)}$ . We choose $\mathrm{d}(G, G')$ in this work for convenience and mathematical tractability, but leave for future work the study of stability relative to $d_{\mathrm{GH}}^s$. Finally,

---

[1]In the sense that it defines a weaker topology, i.e., $\lim_{m \to \infty} \mathrm{d}^s(G, G_m) \to 0 \Rightarrow \lim_{m \to \infty} \mathrm{d}^{s'}(G, G_m) = 0$ for $s' > s$, but not vice-versa.

we consider for simplicity only the case where the sizes of $G$ and $G'$ are equal, but definition (3.1) can be naturally extended to compare variable-sized graphs by replacing permutations by soft-correspondences (see Bronstein et al., 2010).

### 3.4 PROBLEM STATEMENT

Our goal is to build a stable and rich representation $\Phi_G(\mathbf{x})$. The stability property is stated in terms of the diffusion metric above: For a chosen diffusion time $s$, $\forall\, \mathbf{x} \in \mathbb{R}^n$, $G = (V, E, W), G' = (V', E', W')$ with $|V| = |V'| = n$, we want

$$\|\Phi_G(\mathbf{x}) - \Phi_{G'}(\mathbf{x})\| \lesssim \|\mathbf{x}\| \mathrm{d}^s(G, G')\,. \tag{5}$$

This representation can be used to model both signals and domains, or just domains $G$, by considering a prespecified $\mathbf{x} = f(G)$, such as the degree, or by marginalizing from an exchangeable distribution, $\Phi_G = \mathbb{E}_{\mathbf{x} \sim Q} \Phi_G(\mathbf{x})$.

The motivation of (5) is two-fold: On the one hand, we are interested in applications where the signal of interest may be measured in dynamic environments that modify the domain, e.g. in measuring brain signals across different individuals. On the other hand, in other applications, such as building generative models for graphs, we may be interested in representing the domain $G$ itself. A representation from the adjacency matrix of $G$ needs to build invariance to node permutations, while capturing enough discriminative information to separate different graphs. In particular, and similarly as with Gromov-Hausdorff distances, the definition of $\mathrm{d}(G, G')$ involves a matching problem between two kernel matrices, which defines an NP-hard combinatorial problem. This further motivates the need for efficient representations of graphs $\Phi_G$ that can efficiently tell apart two graphs, and such that $\ell(\theta) = \|\Phi_G - \Phi_{G(\theta)}\|$ can be used as a differentiable loss for training generative models.

## 4 GRAPH DIFFUSION SCATTERING

Let $T$ be a lazy diffusion operator associated with a graph $G$ of size $n$ such as those described in Section 3.3. Following Coifman & Maggioni (2006), we construct a family of multiscale filters by exploiting the powers of the diffusion operator $T^{2^j}$. We define

$$\psi_0 := I - T\,, \ \psi_j := T^{2^{j-1}}(I - T^{2^{j-1}}) = T^{2^{j-1}} - T^{2^j}\,, (j > 0)\,. \tag{6}$$

This corresponds to a graph wavelet filter bank with optimal spatial localization. Graph diffusion wavelets are localized both in space and frequency, and favor a spatial localization, since they can be obtained with only two *filter coefficients*, namely $h_0 = 1$ for diffusion $T^{2^{j-1}}$ and $h_1 = -1$ for diffusion $T^{2^j}$. The finest scale $\psi_0$ corresponds to one half of the normalized Laplacian operator $\psi_0 = (1/2)\Delta = 1/2(I - D^{-1/2}WD^{-1/2})$, here seen as a temporal difference in a diffusion process, seeing each diffusion step (each multiplication by $\Delta$) as a time step. The coarser scales $\psi_j$ capture temporal differences at increasingly spaced diffusion times. For $j = 0, \ldots, J_n - 1$, we consider the linear operator

$$\begin{aligned} \mathbf{\Psi} : L^2(G) &\rightarrow (L^2(G))^{J_n} \\ \mathbf{x} &\mapsto (\psi_j \mathbf{x})_{j=0,\ldots,J_n-1}\,, \end{aligned} \tag{7}$$

which is the analog of the wavelet filter bank in the Euclidean domain. Whereas several other options exist to define graph wavelet decompositions (Rustamov & Guibas, 2013; Gavish et al., 2010), and GNN designs that favor frequency localization, such as Cayley filters (Levie et al., 2019), we consider here wavelets that can be expressed with few diffusion terms, favoring spatial over frequential localization, for stability reasons that will become apparent next. We choose dyadic scales for convenience, but the construction is analogous if one replaces scales $2^j$ by $\lceil \gamma^j \rceil$ for any $\gamma > 1$ in (6).

If the graph $G$ exhibits a *spectral gap*, i.e., $\beta_G = \sup_{i=1,\ldots n-1} |\lambda_i| < 1$, the following proposition proves that the linear operator $\mathbf{\Psi}$ defines a stable frame.

**Proposition 4.1.** *For each $n$, let $\mathbf{\Psi}$ define the diffusion wavelet decomposition (7) and assume $\beta_G < 1$. Then there exists a constant $0 < C(\beta)$ depending only on $\beta$ such that for any $\mathbf{x} \in \mathbb{R}^n$*

*satisfying* $\langle \mathbf{x}, \mathbf{v} \rangle = 0$,

$$C(\beta)\|\mathbf{x}\|^2 \le \sum_{j=0}^{J_n-1} \|\psi_j \mathbf{x}\|^2 \le \|\mathbf{x}\|^2 \ . \tag{8}$$

This proposition thus provides the Littlewood-Paley bounds of $\mathbf{\Psi}$, which control the ability of the filter bank to capture and amplify the signal $\mathbf{x}$ along each 'frequency' (i.e. the ability of the filter to increase or decrease the energy of the representation, relative to the energy of the $\mathbf{x}$). We note that diffusion wavelets are neither unitary nor analytic and therefore do not preserve energy. However, the frame bounds in Proposition 4.1 provide lower bounds on the energy lost, such that the smaller $1 - \beta$ is, the less "unitary" our diffusion wavelets are. It also informs us about how the spectral gap $\beta$ determines the appropriate diffusion *scale* $J$: The maximum of $p(u) = (u^r - u^{2r})^2$ is at $u = 2^{-1/r}$, thus the cutoff $r_*$ should align with $\beta$ as $r_* = \frac{-1}{\log_2 \beta}$, since larger values of $r$ capture energy in a spectral range where the graph has no information. Therefore, the maximum scale can be adjusted as $J = \lceil 1 + \log_2 r_* \rceil = 1 + \left\lceil \log_2 \left( \frac{-1}{\log_2 \beta} \right) \right\rceil$.

Recall that the Euclidean Scattering transform is constructed by cascading three building blocks: a wavelet decomposition operator, a pointwise modulus activation function, and an averaging operator. Following the Euclidean scattering, given a graph $G$ and $\mathbf{x} \in L^2(G)$, we define an analogous Diffusion Scattering transform $\Phi_G(\mathbf{x})$ by cascading three building blocks: the Wavelet decomposition operator $\mathbf{\Psi}$, a pointwise activation function $\rho$, and an average operator $U$ which extracts the average over the domain. The average over a domain can be interpreted as the diffusion at infinite time, thus $U\mathbf{x} = \lim_{t \to \infty} T^t \mathbf{x} = \langle \mathbf{v}^\mathsf{T}, \mathbf{x} \rangle$. More specifically, we consider a first layer transformation given by

$$\phi_1(G, \mathbf{x}) = U\rho\mathbf{\Psi}\mathbf{x} = \{U\rho\psi_j \mathbf{x}\}_{0 \le j \le J_n-1} \,,, \tag{9}$$

followed by second order coefficients

$$\phi_2(G, \mathbf{x}) = U\rho\mathbf{\Psi}\rho\mathbf{\Psi}\mathbf{x} = \{U\rho\psi_{j_2}\rho\psi_{j_1}\mathbf{x}\}_{0 \le j_1, j_2 \le J_n-1} \,,, \tag{10}$$

and so on. The representation obtained from $m$ layers of such transformation is thus

$$\Phi_G(\mathbf{x}) = \{U x, \phi_1(G, \mathbf{x}), \dots, \phi_{m-1}(G, \mathbf{x})\} = \{U(\rho\mathbf{\Psi})^k \mathbf{x} \,;\, k = 0, \dots, m-1\} \ . \tag{11}$$

## 5 STABILITY OF GRAPH DIFFUSION SCATTERING

### 5.1 STABILITY AND EQUIVARIANCE OF DIFFUSION WAVELETS

Given two graphs $G, G'$ of size $n$ and a signal $\mathbf{x} \in \mathbb{R}^n$, our objective is to bound $\|\Phi_G(\mathbf{x}) - \Phi_{G'}(\mathbf{x})\|$ in terms of $\mathrm{d}(G, G')$. Let $\pi_*$ the permutation minimising the distortion between $G$ and $G'$ in (4). Since all operations in $\Phi$ are either equivariant or invariant with respect to permutations, we can assume w.l.o.g. that $\pi = 1$, so that the diffusion distance can be directly computed by comparing nodes with the given order. A key property of $G$ that drives the stability of the diffusion scattering is given by its *spectral gap* $1 - \beta_G = 1 - \sup_{i=1,\dots n-1} |\lambda_i| \ge 0$. In the following, we use $\ell_2$ operator norms, unless stated otherwise.

**Lemma 5.1.** *Assume* $\beta := \max(\beta_G, \beta_{G'}) < 1$. *Then*

$$\inf_{\Pi \in \Pi_n} \|\mathbf{\Psi}_G - \Pi\mathbf{\Psi}_{G'}\Pi^\mathsf{T}\| \le 2\mathrm{d}(G, G')\sqrt{\frac{\beta^2(1+\beta^2)}{(1-\beta^2)^3}} \ . \tag{12}$$

*Remark:* If diffusion distance is measured at time different from $s = 1/2$, the stability bound would be modified due to scales $j$ such that $2^j < s$.

The following lemma studies the stability of the low-pass operator $U$ with respect to graph perturbations.

**Lemma 5.2.** *Let* $G, G'$ *be two graphs with same size, denote by* $\mathbf{v}$ *and* $\mathbf{v}'$ *their respective squared-root degree vectors, and by* $\beta, \beta'$ *their spectral gap. Then*

$$\inf_{\Pi \in \Pi_n} \|\mathbf{v} - \Pi\mathbf{v}'\|^2 \le 2\frac{\mathrm{d}(G, G')}{1 - \min(\beta, \beta')} \ . \tag{13}$$

**Spectral Gap asymptotic behavior** Lemmas 5.1 and 5.2 leverage the spectral gap of the lazy diffusion operator associated with $G$. In some cases, such as regular graphs, the spectral gap vanishes asymptotically as $n \to \infty$, thus degrading the upper bound asymptotically. Improving the bound by leveraging other properties of the graph (such as regular degree distribution) is an important open task.

## 5.2 STABILITY AND INVARIANCE OF DIFFUSION SCATTERING

The scattering transform coefficients $\Phi_G(\mathbf{x})$ obtained after $m$ layers are given by equation 11, for low-pass operator $U$ such that $U\mathbf{x} = \langle \mathbf{v}, \mathbf{x} \rangle$ so that $U = \mathbf{v}^\mathsf{T}$.

From Lemma 5.1 we have that, $\|\mathbf{\Psi}_G - \mathbf{\Psi}_{G'}\| \leq \varepsilon_{\mathbf{\Psi}} = 2\mathrm{d}(G, G')\sqrt{\beta^2(1 + \beta^2)/(1 - \beta^2)^3}$. We also know, from Proposition 4.1 that $\mathbf{\Psi}$ conforms a frame, i.e. $C(\beta)\|\mathbf{x}\|^2 \leq \|\mathbf{\Psi x}\|^2 \leq \|\mathbf{x}\|^2$ for known constant $C(\beta)$ given in Prop. 4.1. Additionally, from Lemma 5.2 we get that $\|U_G - U_{G'}\| \leq \varepsilon_U = 2\mathrm{d}(G, G')/(1 - \min(\beta, \beta'))$.

The objective now is to prove stability of the scattering coefficients $\Phi_G(\mathbf{x})$, that is, to prove that

$$\|\Phi_G(\mathbf{x}) - \Phi_{G'}(\mathbf{x})\| \lesssim \mathrm{d}(G, G')\|\mathbf{x}\|. \tag{14}$$

This is captured in the following Theorem:

**Theorem 5.3.** *Let $G, G'$ be two graphs and let $\mathrm{d}(G, G')$ be their distance measured as in equation 4. Let $T_G$ and $T_{G'}$ be the respective diffusion operators. Denote by $U_G$, $\rho_G$ and $\mathbf{\Psi}_G$ and by $U_{G'}$, $\rho_{G'}$ and $\mathbf{\Psi}_{G'}$ the low pass operator, pointwise nonlinearity and the wavelet filter bank used on the scattering transform defined on each graph, respectively, cf. equation 11. Assume $\rho_G = \rho_{G'}$ and that $\rho_G$ is non-expansive. Let $\beta_- = \min(\beta_G, \beta_{G'})$, $\beta_+ = \max(\beta_G, \beta_{G'})$ and assume $\beta_+ < 1$. Then, we have that, for each $k = 0, \ldots, m - 1$, the following holds*

$$\|U_G(\rho_G\mathbf{\Psi}_G)^k - U_{G'}(\rho_{G'}\mathbf{\Psi}_{G'})^k\| \leq \left(\frac{2}{1 - \beta_-}\mathrm{d}(G, G')\right)^{1/2} + k\sqrt{\frac{\beta_+^2(1 + \beta_+^2)}{(1 - \beta_+^2)^3}}\mathrm{d}(G, G'). \tag{15}$$

Defining $\|\Phi_G(\mathbf{x})\|^2 = \sum_{k=0}^{m-1} \|U_G(\rho_G\Psi_G)^k\|^2$ analogously to Bruna & Mallat (2013), it is straightforward to compute the stability bound on the scattering coefficients as follows.

**Corollary 5.4.** *In the context of Theorem 5.3, let $\mathbf{x} \in \mathbb{R}^n$ and let $\Phi_G(\mathbf{x})$ be the scattering coefficients computed by means of equation 11 on graph $G$ after $m$ layers, and let $\Phi_{G'}(\mathbf{x})$ be the corresponding coefficients on graph $G'$. Then,*

$$\|\Phi_G(\mathbf{x}) - \Phi_{G'}(\mathbf{x})\|^2 \leq \sum_{k=0}^{m-1} \left[\left(\frac{2}{1 - \beta_-}\mathrm{d}(G, G')\right)^{1/2} + k\sqrt{\frac{\beta_+^2(1 + \beta_+^2)}{(1 - \beta_+^2)^3}}\mathrm{d}(G, G')\right]^2 \|\mathbf{x}\|^2 \tag{16}$$

$$\|\Phi_G(\mathbf{x}) - \Phi_{G'}(\mathbf{x})\| \lesssim m^{1/2}\mathrm{d}^{1/2}(G, G')\|\mathbf{x}\| \qquad \text{if } \mathrm{d}(G, G') \ll 1.$$

Corollary 5.4 satisfies equation 5. It also shows that the closer the graphs are in terms of the diffusion metric, the closer their scattering representations will be. The constant is given by topological properties, the spectral gaps of $G$ and $G'$, as well as design parameters, the number of layers $m$. We observe that the stability bound grows the smaller the spectral gap is and also as more layers are considered. The spectral gap is tightly linked with diffusion processes on graphs, and thus it does emerge from the choice of a diffusion metric. Graphs with values of beta closer to 1, exhibit weaker diffusion paths, and thus a small perturbation on the edges of these graphs would lead to a larger diffusion distance. The contrary holds as well. In other words, the tolerance of the graph to edge perturbations (i.e., $\mathrm{d}(G, G)$ being small) depends on the spectral gap of the graph. We also note that, as stated at the end of Section 5.1, the spectral gap appears in our upper bounds, but it is not necessarily sharp. In particular, the spectral gap is a poor indication of stability in regular graphs, and we believe our bound can be improved by leveraging structural properties of regular domains.

Finally, we note that the size of the graphs impacts the stability result inasmuch as it impacts the distance measure $\mathrm{d}(G, G')$. This is expected, since graphs of different size can be compared, as mentioned in Section 3.3. Different from Zou & Lerman (2018), our focus is on obtaining graph wavelet banks that are localized in the graph domain to improve computational efficiency as discussed in Defferrard et al. (2016). We also notice that the scattering transform in Zou & Lerman

(2018) is stable with respect to a graph measure that depends on the spectrum of the graph through both eigenvectors and eigenvalues. More specifically, it is required that the spectrum gets concentrated as the graphs grow. However, in general, it is not straightforward to relate the topological structure of the graph with its spectral properties.

As mentioned in Section 3.3, the stability is computed with a metric $d(G, G')$ which is stronger than what could be hoped for. Our metric is permutation-invariant, in analogy with the rigid translation invariance in the Euclidean case, and stable to small perturbations around permutations. The extension of (16) to weaker metrics, using e.g. multiscale deformations, is left for future work.

## 5.3 From Diffusion Scattering to Diffusion GNNs

By denoting $T_j = T^{2^j}$, observe that one can approximate the diffusion wavelets from (6) as a cascade of low-pass diffusions followed by a high-pass filter at resolution $2^j$:

$$\psi_j = T_{j-1}(I - T_{j-1}) \approx T^{\sum_{j' < j-1} 2^{j'}}(I - T_{j-1}) = \left( \prod_{j' < j-1} T_{j'} \right) (I - T_{j-1}).$$

This pyramidal structure of multi-resolution analysis wavelets — in which each layer now corresponds to a different scale, shows that the diffusion scattering is a particular instance of GNNs where each layer $j$ is generated by the pair of operators $\{I, T_{j-1}\}$. If $\mathbf{x}^{(j)} \in \mathbb{R}^{n \times d_j}$ denotes the feature representation at layer $j$ using $d_j$ feature maps per node, the corresponding update is given by

$$\mathbf{x}^{(j+1)} = \rho \left( \mathbf{x}^{(j)} \theta_1^{(j)} + T_{j-1} \mathbf{x}^{(j)} \theta_2^{(j)} \right),$$ (17)

where $\theta_1^{(j)}, \theta_2^{(j)}$ are $d_j \times d_{j+1}$ weight matrices. In this case, a simple modification of the previous theorem shows that the resulting GNN representation $\Phi_G(\mathbf{x}, \Theta)$, $\Theta = (\theta_1^{(j)}, \theta_2^{(j)})_{j \leq J}$ is also stable with respect to $d(G, G')$, albeit this time the constants are parameter-dependent:

**Corollary 5.5.** *The $J$ layer GNN with parameters $\Theta = (\theta_1^{(j)}, \theta_2^{(j)})_{j \leq J}$ satisfies*

$$\|\Phi_G(\mathbf{x}, \Theta) - \Phi_{G'}(\mathbf{x}, \Theta)\| \leq d(G, G') \frac{\|\mathbf{x}\|}{1 - \beta} \left[ \prod_{j \leq J} (1 + \|\theta_1^{(j)}\| + \|\theta_2^{(j)}\|) \right]^2.$$ (18)

This bound is thus learning-agnostic and is proved by elementary application of the diffusion distance definition. An interesting question left for future work is to relate such stability to gradient descent optimization biases, similarly as in (Gunasekar et al., 2018; Wei et al., 2018), which could provide stability certificates for learnt GNN representations.

## 6 Numerical Experiments

In this section, we first show empirically the dependence of the stability result with respect to the spectral gap, and then we illustrate the discriminative power of the diffusion scattering transform in two different classification tasks; namely, the problems of authorship attribution and source localization.

Consider a small-world graph $G$ with $N = 200$ nodes, edge probability $p_{SW}$ and rewiring probability $q_{SW} = 0.1$. Let $\mathbf{x} \sim \mathcal{N}(0, \mathbf{I})$ be a random graph signal defined on top of $G$ and $\Phi_G(\mathbf{x})$ the corresponding scattering transform. Let $G'$ be another realization of the small-world graph, and let $\Phi_{G'}(\mathbf{x})$ be the scattering representation of the same graph signal $\mathbf{x}$ but on the different support $G'$. We can then proceed to compute $\|\Phi_G(\mathbf{x}) - \Phi_{G'}(\mathbf{x})\|$. By changing the value of $p_{SW}$ we can change value of the spectral gap $\beta$ and study the dependence of the difference in representations as a function of the spectral gap. Results shown in Fig. 1a are obtained by varying $p_{SW}$ from 0.1 to 0.9. For each value of $p_{SW}$ we generate one graph $G$ and 50 different graphs $G'$; and for each graph $G'$ we compute $\|\Phi_G(\mathbf{x}) - \Phi_{G'}(\mathbf{x})\|$ for $1,000$ different graph signal realizations $\mathbf{x}$. The average across all signal realizations is considered the estimate of the representation difference, and then the mean as well as the variance across all graphs are computed and shown in the figure (error bars).

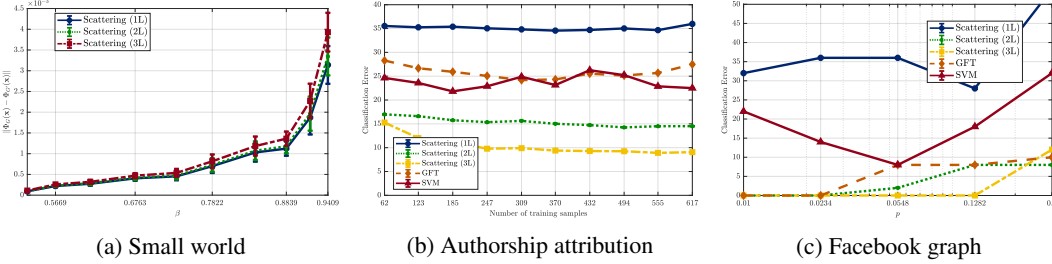

(a) Small world          (b) Authorship attribution          (c) Facebook graph

Figure 1. (a) Difference in representation between the signal defined on the original graph $G$ and on the deformed graph $G'$ as a function of the spectral gap $\beta$. (b)-(c) Classification error percentage as a function of perturbation for the authorship attribution and the Facebook graph, respectively.

Fig. 1a shows the average difference $\|\Phi_G(\mathbf{x}) - \Phi_G(\mathbf{x}')\|$ as a function of the spectral gap (changing $p_{\mathrm{SW}}$ from 0.1 to 0.9 led to values of spectral gap between 0.5 and close to 1). First and foremost we observe that, indeed, as $\beta$ reaches one, the stability result gets worse and the representation difference increases. We also observe that, for deeper scattering representations, the difference also gets worse, although it is not a linear behaviour as predicted in equation 16, which suggest that the bound is not tight.

For classifying we train a SVM linear model fed by features obtained from different representations. We thus compare with two non-trainable linear representations of the data: a data-based method (using the graph signals to feed the classifier) and a graph-based method (obtaining the GFT coefficients as features for the data). Additionally, we consider the graph scattering with varying depth to analyze the richness of the representation. Our aim is mainly to illustrate that the scattering representation is rich enough, relative to linear representations, and is stable to graph deformations.

First, we consider the problem of authorship attribution where the main task is to determine if a given text was written by a certain author. We construct author profiles by means of word adjacency networks (WAN). This WAN acts as the underlying graph support for the graph signal representing the word count (bag-of-words) of the target text of unknown authorship. Intuitively, the choice of words of the target text should reflect the pairwise connections in the WAN, see Segarra et al. (2015) for detailed construction of WANs. In particular, we consider all works by Jane Austen. To illustrate the stability result, we construct a WAN with 188 nodes (functional words) using a varying number of texts to form the training set, obtaining an array of graphs that are similar but not exactly the same. For the test set, we include 154 excerpts by Jane Austen and 154 excerpts written by other contemporary authors, totaling 308 data points. Fig. 1b shows classification error as a function of the number of training samples used. We observe that graph scattering transforms monotonically improve while considering more training data, whereas other methods vary more erratically, showing their lack of stability (their representations vary more wildly when the underlying graph support changes). This shows that scattering diffusion transforms strike a good balance between stability and discriminative power.

For the second task, let $G$ be a 234-node graph modeling real-world Facebook interactions (McAuley & Leskovec, 2012). In the source localization problem, we observe a diffusion process after some unknown time $t$, that originated at some unknown node $i$, i.e. we observe $\mathbf{x} = W^t \delta_i$, where $\delta_i$ is the signal with all zeros except a 1 on node $i$. The objective is to determine which community the source node $i$ belongs to. These signals can be used to model rumors that percolate through the social network by interaction between users and the objective is to determine which user group generated said rumor (or initiated a discussion on some topic). We generate a training sample of size $2,000$, for nodes $i$ chosen at random and diffusion times $t$ chosen as random as well. The GFT is computed by projecting on the eigenbasis of the operator $T$. We note that, to avoid numerical instabilities, the diffusion is carried out using the normalized operator $(W/\lambda_{\max}(W))$ and $t \leq t_{\max} = 20$. The representation coefficients (graph signals, GFT or scattering coefficients) obtained from this set are used to train different linear SVMs to perform classification. For the test set, we draw 200 new signals. We compute the classification errors on the test set as a measure of usefulness of the obtained representations. Results are presented in Fig. 1c, where perturbations are illustrated by dropping edges with probability $p$ (adding or removing friends in Facebook). Again, it is observed that the

scattering representation exhibits lower variations when the underlying graph changes, compared to the linear approaches.

Finally, to remark the discriminative power of the scattering representation, we observe that as the graph scattering grows deeper, the obtained features help in more accurate classification. We remark that in regimes with sufficient labeled examples, trainable GNN architectures will generally outperform scattering-based representations.

## 7 CONCLUSIONS

In this work we addressed the problem of stability of graph representations. We designed a scattering transform of graph signals using diffusion wavelets and we proved that this transform is stable under deformations of the underlying graph support. More specifically, we showed that the scattering transform of a graph signal supported on two different graphs is proportional to the diffusion distance between those graphs. As a byproduct of our analysis, we obtain stability bounds for Graph Neural Networks generated by diffusion operators. Additionally, we showed that the resulting descriptions are also rich enough to be able to adequately classify plays by author in the context of authorship attribution, and identify the community origin of a signal in a source localization problem.

That said, there are a number of directions to build upon from these results. First, our stability bounds depend on the spectral gap of the graph diffusion. Although lazy diffusion prevents this spectral gap to vanish, as the size of the graph increases we generally do not have a tight bound, as illustrated by regular graphs. An important direction of future research is thus to develop stability bounds which are robust to vanishing spectral gaps. Next, and related to this first point, we are working on extending the analysis to broader families of wavelet decompositions on graphs and their corresponding graph neural network versions, including stability with respect to the Gromov-Hausdorff metric, which can be achieved by using graph wavelet filter banks that achieve bounds analogous to those in Lemmas 5.1 and 5.2.

### ACKNOWLEDGMENTS

Work supported by NSF CCF 1717120, US ARO W911NF1710438, ARL DCIST CRA W911NF-17-2-0181, ISTC-WAS and Intel AI DevCloud.

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

## A    PROOF OF PROPOSITION 4.1

Since all operators $\psi_j$ are polynomials of the diffusion $T$, they all diagonalise in the same basis. Let $T = V\Lambda V^\mathsf{T}$, where $V^\mathsf{T}V = I$ contains the eigenvectors of $T$ and $\Lambda = \mathrm{diag}(\lambda_0, \ldots, \lambda_{n-1})$ its eigenvalues. The frame bounds $C_1, C_2$ are obtained by evaluating $\|\Psi\mathbf{x}\|^2$ for $\mathbf{x} = \mathbf{v}_i, i = 1, \ldots, n-1$, since $\mathbf{v}_0$ corresponds to the square-root degree vector and $\mathbf{x}$ is by assumption orthogonal to $\mathbf{v}_0$.

We verify that the spectrum of $\psi_j$ is given by $(p_j(\lambda_0), \ldots, p_j(\lambda_{n-1}))$, where $p_j(x) = x^{2^{j-1}} - x^{2^j}$ for $j > 0$ and $p_0(x) = 1 - x$. Denote by $Q_J(x) = \sum_{j=0}^{J-1} p_j(x)^2$. It follows from the definition that $\|\Psi\mathbf{v}_i\|^2 = Q_J(\lambda_i)$ for $i = 1, \ldots, n-1$ and therefore

$$C_1 = \min_{x \in (0,\beta)} Q_J(x), \ C_2 = \max_{x \in (0,\beta)} Q_J(x). \tag{19}$$

We check that $C_1 \geq \min_{x \in (0,\beta)} p_0(x)^2 = (1 - \beta)^2$ and $C_2 = Q_J(0) = 1$. Indeed, denote by $Q(x) = \sum_{j=0}^{\infty} p_j(x)^2$. One easily verifies that $Q(x)$ is continuous in $[0, 1)$ since it is bounded by a geometric series. Also, observe that $Q(x) = (1-x)^2 + \sum_{j>0}(x^{2^{j-1}} - x^{2^j})^2$ satisfies the recurrence

$$Q(x^2) = Q(x) + 2x(1 - x)^2 \geq Q(x)$$

since $x \in [0, 1)$. By continuity it thus follows that

$$\sup_{x \in [0,1)} Q_J(x) \leq \sup_{x \in [0,1)} Q(x) = \lim_{x \to 0} Q(x) = Q(0) = 1 \ \Box.$$

## B    PROOF OF LEMMA 5.1

By definition $\|\Psi_G - \Psi_{G'}\|^2 = \|[\Psi_G - \Psi_{G'}]^*[\Psi_G - \Psi_{G'}]\|$ and

$$[\Psi_G - \Psi_{G'}]^*[\Psi_G - \Psi_{G'}] = \sum_{j=0}^{J_n-1} (\psi_j(G) - \psi_j(G'))^*(\psi_j(G) - \psi_j(G')). \tag{20}$$

Since $T_G$ and $T_{G'}$ are self-adjoint, so are $\psi_j(G)$ and $\psi_j(G')$. From the triangular inequality, we thus obtain

$$\|\Psi_G - \Psi_{G'}\|^2 \leq \sum_j \|\psi_j(G) - \psi_j(G')\|^2. \tag{21}$$

Denote by $\mathbf{v}$ the eigenvector associated with $\lambda_0 = 1$, $[\mathbf{v}]_i = (d_i / \sum_{j=1}^n d_j)^{1/2}$. Since $\beta < 1$, we can write $T = \mathbf{v}\mathbf{v}^\mathsf{T} + \overline{T}$ with $\|\overline{T}\| < 1$ and $\mathbf{v} \in \mathrm{Null}(\overline{T})$. It follows by induction that $T^r = \mathbf{v}\mathbf{v}^\mathsf{T} + \overline{T}^r$, and hence

$$\psi_j(G) = T_G^{2^{j-1}} - T_G^{2^j} = \overline{T}_G^{2^{j-1}} - \overline{T}_G^{2^j},$$

and equivalently for $G'$, resulting in

$$\|\psi_j(G) - \psi_j(G')\|^2 \leq 2\left(\|\overline{T}_G^{2^{j-1}} - \overline{T}_{G'}^{2^{j-1}}\|^2 + \|\overline{T}_G^{2^j} - \overline{T}_{G'}^{2^j}\|^2\right),$$

and thus

$$\|\Psi_G - \Psi_{G'}\|^2 \leq 4\sum_j \|\overline{T}_G^{2^j} - \overline{T}_{G'}^{2^j}\|^2. \tag{22}$$

Let us show that for matrices $A$ and $B$ such that $\beta = \max(\|A\|, \|B\|) < 1$ and $r \in \mathbb{N}$, one has

$$\|A^r - B^r\| \leq r\beta^{r-1}\|A - B\|. \tag{23}$$

Indeed, by noting $g(t) = (tB + (1 - t)A)^r$, we have

$$\|A^r - B^r\| = \|g(1) - g(0)\| = \left\|\int_0^1 g'(t)dt\right\| \leq \int_0^1 \|g'(t)\|dt \leq \sup_{t \in (0,1)} \|g'(t)\|.$$

By noting $A_t = tB + (1 - t)A$ we verify that

$$g'(t) = \sum_{l=0}^{r-1} A_t^l (B - A) A_t^{r-l-1} \,,$$

which results in $\|g'(t)\| \leq r\beta^{r-1}\|B - A\|$, proving (23).

By plugging (23) into (22) we thus obtain

$$
\begin{aligned}
\|\mathbf{\Psi}_G - \mathbf{\Psi}_{G'}\|^2 &\leq 4\|\overline{T}_G - \overline{T}_{G'}\|^2 \sum_j 2^{2j} \beta^{2^{j+1}} && (24) \\
&\leq 4\|\overline{T}_G - \overline{T}_{G'}\|^2 \sum_t t^2 (\beta^2)^t \\
&\leq 4\|\overline{T}_G - \overline{T}_{G'}\|^2 \frac{\beta^2(1+\beta^2)}{(1-\beta^2)^3} \,,
\end{aligned}
$$

which yields $\|\mathbf{\Psi}_G - \mathbf{\Psi}_{G'}\| \leq 2\|\overline{T}_G - \overline{T}_{G'}\|\sqrt{\frac{\beta^2(1+\beta^2)}{(1-\beta^2)^3}}$. Finally, we observe that $\|\overline{T}_G - \overline{T}_{G'}\| = \|T_G - T_{G'}\|$, which proves (12) as claimed. $\square$

## C  PROOF OF LEMMA 5.2

Without loss of generality, assume that the node assignment that minimizes $\|T_G - \Pi T_{G'}\Pi\|^\mathsf{T}$ is the identity. We need to bound the leading eigenvectors of two symmetric matrices $T_G$ and $T_{G'}$ with a spectral gap. As before, let $T_G = \mathbf{v}\mathbf{v}^\mathsf{T} + \overline{T}_G$ and $T_{G'} = \mathbf{v}'(\mathbf{v}')^\mathsf{T} + \overline{T}_{G'}$. Let $\alpha = \langle \mathbf{v}, \mathbf{v}' \rangle \geq 0$ since both are non-negative vectors. Denote by $E = T_G - T_{G'}$ and $\overline{E} = \overline{T}_G - \overline{T}_{G'}$. Then $E = \mathbf{v}\mathbf{v}^\mathsf{T} - \mathbf{v}'(\mathbf{v}')^\mathsf{T} + \overline{E}$. Hence $E\mathbf{v} = \mathbf{v} - \mathbf{v}'\alpha - \overline{T}_{G'}(\mathbf{v} - \alpha\mathbf{v}')$, so

$$(\mathbf{I} - \overline{T}_{G'})(\mathbf{v} - \mathbf{v}'\alpha) = E\mathbf{v}$$

Since $\|E\| \leq \mathrm{d}(G, G')$ and $\|\overline{T}_{G'}\| < \beta'$, we have

$$(1 - \alpha)(1 - \beta') \leq \|(\mathbf{I} - \overline{T}_{G'})(\mathbf{v} - \mathbf{v}'\alpha)\| \leq \mathrm{d}(G, G') \,,$$

so

$$1 - \alpha \leq \frac{\mathrm{d}(G, G')}{1 - \beta'} \,.$$

Finally, since $\|\mathbf{v} - \mathbf{v}'\| = \sqrt{2 - 2\alpha}$, we have

$$\|\mathbf{v} - \mathbf{v}'\|^2 \leq 2\frac{\mathrm{d}(G, G')}{1 - \beta'} \,. \tag{25}$$

Since we are free to swap the role of $\mathbf{v}$ and $\mathbf{v}'$, the result follows. $\square$

## D  PROOF OF THEOREM 5.3

First, note that $\rho_G = \rho_{G'} = \rho$ since it is a pointwise nonlinearity (an absolute value), and is independent of the graph topology. Now, let's start with $k = 0$. In this case, we get $\|U_G\mathbf{x} - U_{G'}\mathbf{x}\|$ which is immediately bounded by Lemma 5.2 satisfying equation 15.

For $k = 1$ we have

$$
\begin{aligned}
\|U_G\rho\mathbf{\Psi}_G\mathbf{x} - U_{G'}\rho\mathbf{\Psi}_{G'}\mathbf{x}\| &= \|U_G\rho\mathbf{\Psi}_G\mathbf{x} - U_{G'}\rho\mathbf{\Psi}_G\mathbf{x} + U_{G'}\rho\mathbf{\Psi}_G\mathbf{x} - U_{G'}\rho\mathbf{\Psi}_{G'}\mathbf{x}\| && (26) \\
&\leq \|(U_G - U_{G'})\rho\mathbf{\Psi}_G\mathbf{x}\| + \|U_{G'}\rho((\mathbf{\Psi}_G - \mathbf{\Psi}_{G'})\mathbf{x})\| && (27)
\end{aligned}
$$

where the triangular inequality of the norm was used, together with the fact that $\|\rho\mathbf{u} - \rho\mathbf{u}'\| \leq \|\rho(\mathbf{u} - \mathbf{u}')\|$ for any real vector $\mathbf{u}$ since $\rho$ is the pointwise absolute value. Using the submultiplicativity of the operator norm, we get

$$\|U_G\rho\mathbf{\Psi}_G\mathbf{x} - U_{G'}\rho\mathbf{\Psi}_{G'}\mathbf{x}\| \leq \|U_G - U_{G'}\|\|\rho\|\|\mathbf{\Psi}_G\|\|\mathbf{x}\| + \|U_{G'}\|\|\rho\|\|\mathbf{\Psi}_G - \mathbf{\Psi}_{G'}\|\|\mathbf{x}\|. \tag{28}$$

From Lemmas 5.1 and 5.2 we have that $\|\mathbf{\Psi}_G - \mathbf{\Psi}_{G'}\| \le \varepsilon_{\mathbf{\Psi}}$ and $\|U_G - U_{G'}\| \le \varepsilon_U$, and from Proposition 4.1 that $\|\mathbf{\Psi}_G\| \le 1$. Note also that $\|U_{G'}\| = \|U_G\| = 1$ and that $\|\rho\| = 1$. This yields

$$\|U_G \rho \mathbf{\Psi}_G \mathbf{x} - U_{G'} \rho \mathbf{\Psi}_{G'} \mathbf{x}\| \le \varepsilon_U \|\mathbf{x}\| + \varepsilon_{\mathbf{\Psi}} \|\mathbf{x}\|. \tag{29}$$

satisfying equation 15 for $k = 1$.

For $k = 2$, we observe that

$$\|U_G \rho \mathbf{\Psi}_G \rho \mathbf{\Psi}_G \mathbf{x} - U_{G'} \rho \mathbf{\Psi}_{G'} \rho \mathbf{\Psi}_{G'} \mathbf{x}\|$$
$$= \|U_G \rho \mathbf{\Psi}_G \rho \mathbf{\Psi}_G \mathbf{x} - U_{G'} \rho \mathbf{\Psi}_G \rho \mathbf{\Psi}_G \mathbf{x} + U_{G'} \rho \mathbf{\Psi}_G \rho \mathbf{\Psi}_G \mathbf{x} - U_{G'} \rho \mathbf{\Psi}_{G'} \rho \mathbf{\Psi}_{G'} \mathbf{x}\| \tag{30}$$
$$\le \|(U_G - U_{G'}) \rho \mathbf{\Psi}_G \rho \mathbf{\Psi}_G \mathbf{x}\| + \|U_{G'} (\rho \mathbf{\Psi}_G \rho \mathbf{\Psi}_G \mathbf{x} - \rho \mathbf{\Psi}_{G'} \rho \mathbf{\Psi}_{G'} \mathbf{x})\| \tag{31}$$

The first term is bounded in a straightforward fashion by $\|(U_G - U_{G'}) \rho \mathbf{\Psi}_G \rho \mathbf{\Psi}_G \mathbf{x}\| \le \varepsilon_U \|\mathbf{x}\|$ in analogy to the development for $k = 1$. Since $\|U_{G'}\| = 1$, for the second term, we focus on

$$\|\rho \mathbf{\Psi}_G \rho \mathbf{\Psi}_G \mathbf{x} - \rho \mathbf{\Psi}_{G'} \rho \mathbf{\Psi}_{G'} \mathbf{x}\| = \|\rho \mathbf{\Psi}_G \rho \mathbf{\Psi}_G \mathbf{x} - \rho \mathbf{\Psi}_G \rho \mathbf{\Psi}_{G'} \mathbf{x} + \rho \mathbf{\Psi}_G \rho \mathbf{\Psi}_{G'} \mathbf{x} - \rho \mathbf{\Psi}_{G'} \rho \mathbf{\Psi}_{G'} \mathbf{x}\| \tag{32}$$

$$\le \|\rho \mathbf{\Psi}_G \rho \mathbf{\Psi}_G \mathbf{x} - \rho \mathbf{\Psi}_G \rho \mathbf{\Psi}_{G'} \mathbf{x}\| + \|\rho \mathbf{\Psi}_G \rho \mathbf{\Psi}_{G'} \mathbf{x} - \rho \mathbf{\Psi}_{G'} \rho \mathbf{\Psi}_{G'} \mathbf{x}\| \tag{33}$$

We note that, in the first term in equation 33, the first layer induces an error, but after that, the processing is through the same filter banks. So we are basically interested in bounding the propagation of the error induced in the first layer. Applying twice the fact that $\|\rho(\mathbf{u}) - \rho(\mathbf{u}')\| \le \|\rho(\mathbf{u} - \mathbf{u}')\|$ we get

$$\|\rho \mathbf{\Psi}_G \rho \mathbf{\Psi}_G \mathbf{x} - \rho \mathbf{\Psi}_G \rho \mathbf{\Psi}_{G'} \mathbf{x}\| \le \|\rho(\mathbf{\Psi}_G(\rho \mathbf{\Psi}_G \mathbf{x} - \rho \mathbf{\Psi}_{G'} \mathbf{x}))\| \le \|\rho(\mathbf{\Psi}_G \rho((\mathbf{\Psi}_G - \rho \mathbf{\Psi}_{G'}) \mathbf{x}))\|. \tag{34}$$

And following with submultiplicativity of the operator norm,

$$\|\rho \mathbf{\Psi}_G \rho \mathbf{\Psi}_G \mathbf{x} - \rho \mathbf{\Psi}_G \rho \mathbf{\Psi}_{G'} \mathbf{x}\| \le \varepsilon_{\mathbf{\Psi}} \|\mathbf{x}\|. \tag{35}$$

For the second term in equation 33, we see that the first layer applied is the same in both, namely $\rho \mathbf{\Psi}_{G'}$ so there is no error induced. Therefore, we are interested in the error obtained after the first layer, which is precisely the same error obtained for $k = 1$. Therefore,

$$\|\rho \mathbf{\Psi}_G \rho \mathbf{\Psi}_{G'} \mathbf{x} - \rho \mathbf{\Psi}_{G'} \rho \mathbf{\Psi}_{G'} \mathbf{x}\| = \|\rho \mathbf{\Psi}_G \mathbf{x} - \rho \mathbf{\Psi}_{G'} \mathbf{x}\| \le \varepsilon_{\mathbf{\Psi}} \|\mathbf{x}\|. \tag{36}$$

Plugging equation 35 and equation 36 back in equation 31 we get

$$\|U_G \rho \mathbf{\Psi}_G \rho \mathbf{\Psi}_G \mathbf{x} - U_{G'} \rho \mathbf{\Psi}_{G'} \rho \mathbf{\Psi}_{G'} \mathbf{x}\| \le \varepsilon_U \|\mathbf{x}\| + \varepsilon_{\mathbf{\Psi}} \|\mathbf{x}\| + \varepsilon_{\mathbf{\Psi}} \|\mathbf{x}\| \tag{37}$$

satisfying equation 15 for $k = 2$.

For general $k$ we see that we will have a first term that is the error induced by the mismatch on the low pass filter that amounts to $\varepsilon_U$, a second term that accounts for the propagation through $(k - 1)$ equal layers of an initial error, yielding $\varepsilon_{\mathbf{\Psi}}$, and a final third term that is the error induced by the previous layer, $(k - 1)\varepsilon_{\mathbf{\Psi}}$. More formally, assume that equation 15 holds for $k - 1$, implying that

$$\|(\rho \mathbf{\Psi}_G)^{k-1} \mathbf{x} - (\rho \mathbf{\Psi}_{G'})^{k-1} \mathbf{x}\| \le (k - 1)\varepsilon_{\mathbf{\Psi}} \|\mathbf{x}\| \tag{38}$$

Then, for $k$, we can write

$$\|U_G(\rho \mathbf{\Psi}_G)^k \mathbf{x} - U_{G'}(\rho \mathbf{\Psi}_{G'})^k \mathbf{x}\| \le \|(U_G - U_{G'})(\rho \mathbf{\Psi}_G)^k \mathbf{x}\| + \|U_{G'}((\rho \mathbf{\Psi}_G)^k \mathbf{x} - (\rho \mathbf{\Psi}_{G'})^k \mathbf{x})\| \tag{39}$$

Again, the first term we bound it in a straightforward manner using submultiplicativity of the operator norm

$$\|(U_G - U_{G'})(\rho \mathbf{\Psi}_G)^k \mathbf{x}\| \le \varepsilon_U \|\mathbf{x}\|. \tag{40}$$

For the second term, since $\|U_{G'}\| = 1$ we focus on

$$\|(\rho \mathbf{\Psi}_G)^k \mathbf{x} - (\rho \mathbf{\Psi}_{G'})^k \mathbf{x}\| = \|(\rho \mathbf{\Psi}_G)^k \mathbf{x} - (\rho \mathbf{\Psi}_G)^{k-1} \rho \mathbf{\Psi}_{G'} \mathbf{x} + (\rho \mathbf{\Psi}_G)^{k-1} \rho \mathbf{\Psi}_{G'} \mathbf{x} - (\rho \mathbf{\Psi}_{G'})^k \mathbf{x}\| \tag{41}$$

$$\le \|(\rho \mathbf{\Psi}_G)^{k-1} \rho \mathbf{\Psi}_G \mathbf{x} - (\rho \mathbf{\Psi}_G)^{k-1} \rho \mathbf{\Psi}_{G'} \mathbf{x}\| + \|(\rho \mathbf{\Psi}_G)^{k-1} \rho \mathbf{\Psi}_{G'} \mathbf{x} - (\rho \mathbf{\Psi}_{G'})^{k-1} \rho \mathbf{\Psi}_{G'} \mathbf{x}\| \tag{42}$$

The first term in equation 42 computes the propagation in the initial error caused by the first layer. Then, repeatedly applying $\|\rho(\mathbf{u}) - \rho(\mathbf{u}')\| \le \|\rho(\mathbf{u} - \mathbf{u}')\|$ in analogy with $k = 2$ and using submultiplicativity, we get

$$\|(\rho \mathbf{\Psi}_G)^{k-1} \rho \mathbf{\Psi}_G \mathbf{x} - (\rho \mathbf{\Psi}_G)^{k-1} \rho \mathbf{\Psi}_{G'} \mathbf{x}\| \le \varepsilon_{\Psi} \|\mathbf{x}\|. \tag{43}$$

The second term in equation 42 is the bounded by equation 38, since the first layer is exactly the same in this second term. Then, combining equation 43 with equation 38, yields

$$\|(\rho\boldsymbol{\Psi}_G)^k\mathbf{x} - (\rho\boldsymbol{\Psi}_{G'})^k\mathbf{x}\| \le \varepsilon_{\boldsymbol{\Psi}} + (k-1)\varepsilon_{\boldsymbol{\Psi}}\|\mathbf{x}\| = k\varepsilon_{\boldsymbol{\Psi}}\|\mathbf{x}\|. \tag{44}$$

Overall, we get

$$\|U_G(\rho\boldsymbol{\Psi}_G)^k\mathbf{x} - U_{G'}(\rho\boldsymbol{\Psi}_{G'})^k\mathbf{x}\| \le \varepsilon_U\|\mathbf{x}\| + \varepsilon_{\boldsymbol{\Psi}}\|\mathbf{x}\| \tag{45}$$

which satisfies equation 15 for $k$. Finally, since this holds for $k = 2$, the proof is completed by induction. $\square$

## E    PROOF OF COROLLARY 5.4

From Theorem 5.3, we have

$$\|U_G(\rho_G\boldsymbol{\Psi}_G)^k - U_{G'}(\rho_{G'}\boldsymbol{\Psi}_{G'})^k\| \le \left(\frac{2}{1-\beta_-}\mathrm{d}^s(G,G')\right)^{1/2} + k\sqrt{\frac{\beta_+^2(1+\beta_+^2)}{(1-\beta_+^2)^3}}\mathrm{d}(G,G') \tag{46}$$

and, by definition (Bruna & Mallat, 2013, Sec. 3.1),

$$\|\Phi_G(\mathbf{x})\|^2 = \sum_{k=0}^{m-1}\|U_G(\rho_G\boldsymbol{\Psi}_G)^k\mathbf{x}\|^2 \tag{47}$$

so that

$$\|\Phi_G(\mathbf{x}) - \Phi_{G'}(\mathbf{x})\|^2 = \sum_{k=0}^{m-1}\|U_G(\rho_G\boldsymbol{\Psi}_G)^k\mathbf{x} - U_{G'}(\rho_{G'}\boldsymbol{\Psi}_{G'})^k\mathbf{x}\|^2 \tag{48}$$

Then, applying the inequality of Theorem 5.3, we get

$$\|\Phi_G - \Phi_{G'}\|^2 \le \sum_{k=0}^{m-1}\left[\left(\frac{2}{1-\beta_-}\mathrm{d}(G,G')\right)^{1/2} + k\sqrt{\frac{\beta_+^2(1+\beta_+^2)}{(1-\beta_+^2)^3}}\mathrm{d}(G,G')\right]^2 \tag{49}$$

Now, considering each term, such that

$$\|\Phi_G - \Phi_{G'}\|^2 \le \sum_{k=0}^{m-1}\left(\frac{2}{1-\beta_-}\mathrm{d}(G,G')\right) + \sum_{k=0}^{m-1}k^2\frac{\beta_+^2(1+\beta_+^2)}{(1-\beta_+^2)^3}\mathrm{d}^2(G,G') \tag{50}$$

$$+ \sum_{k=0}^{m-1}2^{3/2}k\sqrt{\frac{\beta_+^2(1+\beta_+^2)}{(1-\beta_-)(1-\beta_+^2)^3}}\,\mathrm{d}^{3/2}(G,G') \tag{51}$$

we observe that the second and third term vanish for $\mathrm{d}(G,G') \ll 1$, leaving only the first term, yielding

$$\|\Phi_G - \Phi_{G'}\|^2 \lesssim m\left(\frac{2}{1-\beta_-}\mathrm{d}(G,G')\right). \tag{52}$$

Finally, this leads to the corollary result

$$\|\Phi_G(\mathbf{x}) - \Phi_G(\mathbf{x})\| \lesssim m^{1/2}\,\mathrm{d}^{1/2}(G,G')\|\mathbf{x}\| \text{ if } d(G,G') \ll 1 \tag{53}$$

completing the proof.

## F    PROOF OF COROLLARY 5.5

Let us denote by $e_j = \|x_G^{(j)} - x_{G'}^{(j)}\|$. From (17), from the triangle inequality we verify that

$$e_{j+1} \le a_je_j + b_j \text{ , with}$$

$$a_j = \|\theta_1^{(j)}\| + \|\theta_2^{(j)}\| \text{ , } b_j = \beta^{2^{j-1}}\mathrm{d}(G,G')\|x_{G'}^{(j)}\| \text{ and } e_0 = 0\,.$$

It follows that

$$e_{J+1} \leq \sum_{j \leq J} \left( \prod_{i=j+1}^{J} a_i \right) b_j \leq \left( \prod_{j=1}^{J} (1 + a_j) \right) \left( \sum_{j=1}^{J} b_j \right) , \tag{54}$$

and since

$$b_j = \beta^{2^{j-1}} \mathrm{d}(G, G') \|x_{G'}^{(j)}\| \leq \beta^{2^{j-1}} \mathrm{d}(G, G') \|x\| \left( \prod_{j=1}^{J} (1 + a_j) \right) ,$$

we obtain

$$e_{J+1} \leq \left( \prod_{j=1}^{J} (1 + a_j) \right)^2 \mathrm{d}(G, G') \|x\| \sum_{j} \beta^{2^{j-1}} \leq \left( \prod_{j=1}^{J} (1 + a_j) \right)^2 \mathrm{d}(G, G') \|x\| \frac{1}{1 - \beta} , \tag{55}$$

as desired. $\square$

