# OpenReview forum: "Diffusion Scattering Transforms on Graphs"
_ICLR.cc/2019/Conference_

### Official Review · AnonReviewer1 · 2018-11-02
**interesting study of stability of signal representations on graphs**

**Rating:** 7
**Confidence:** 3

**Review:**

The paper introduces an adaptation of the Scattering transform to signals defined on graphs
by relying on multi-scale diffusion wavelets, and studies a notion of stability of this representation
with respect to changes in the graph structure with an appropriate diffusion metric.

The notion of stability in convolutional networks is an important one, and the proposed notion of stability
with respect to diffusion distances seems like an interesting and relevant way to extend this to signals on graphs.
With this goal in mind, the authors introduce a scattering transform on graphs by relying on diffusion wavelets,
and provide an appropriate study of stability, which seems to highlight relevant properties of the graphs.
The proposed representation seems to provide benefits compared to the previous work of Zou & Lerman,
particularly regarding computational efficiency, as well as stability with respect to a metric that is perhaps more
useful, though there is a dependence on the graph topology through the spectral gaps.
In addition, the experiments on author attribution and source localization suggest that the
resulting representation remains discriminative, in addition to providing stability to changes in graph structure.

I find that these contributions provide an interesting advance in theoretical understanding of graph convolutional networks
from a stability perspective, in addition to introducing a useful non-learned representation,
and am thus in favor of acceptance.

Nevertheless, some parts of paper would benefit from further discussions and more clarity:

- other than empirically, one aspect that's missing compared to the original study of the scattering transform is energy preservation. The authors could at least provide a discussion of whether such a property can be obtained here as well (does it depend on the spectral gap through C(beta)?)

- what is the role of the spectral gap in the stability bounds? is this a drawback of the diffusion metric / choice of wavelets?

- Section 3.2 suggests that metric stability is a good way to characterize stability, by seeing deformations in Euclidian domains as a change to the ground metric. Yet, in Euclidian scattering, the same representation is applied to a deformed signal and the original signal, and stability is measured with the Euclidian metric.
Can the link be made more precise, by explaining what a deformation of a signal would be on a graph, or by applying arguments from the proposed construction to the Euclidian case?

- the paper is heavy on terminology from wavelets and harmonic analysis, a more detailed presentation of diffusion wavelets and related concepts such as localization would be beneficial. Also, it seems like the chosen wavelets in the construction favor spatial over frequential localization - is this due to a trade-off? if so, can it be avoided?


Some more detailed comments:
- Section 2, 'generally far weaker': what is meant by 'weaker'?
- Section 3.3:
  * 'calculus on T': T is used before being defined
  * clarify what norm is used (I assume operator norm?)
  * 'defines a distance', 'stronger than .. GH': this should probably be justified
- Section 4:
  * 'optimal spatial localization', 'temporal difference', 'favoring spatial over frequential localization': these could be clarified
  * 'amplify the signal': what does this mean?
  * the sentence about the choice of the appropriate J is not clear and should be further clarified
- Section 5.1:
  * the sentence about the choice pi/pi* = 1 should be clarified. Also, where is this assumption used?
  * epsilon_psi, epsilon_U should be defined
  * 'given that [..] by definition': this doesn't seem to be defined elsewhere
  * (16): isn't a factor m missing in the first term?

---

> ### Author Response · Authors · 2018-11-12
> **Response to AnonReviewer1: Part 1**
>
> REVIEW: “The paper introduces an adaptation of the Scattering transform to signals defined on graphs by relying on multi-scale diffusion wavelets, and studies a notion of stability of this representation with respect to changes in the graph structure with an appropriate diffusion metric.
>
> The notion of stability in convolutional networks is an important one, and the proposed notion of stability with respect to diffusion distances seems like an interesting and relevant way to extend this to signals on graphs. With this goal in mind, the authors introduce a scattering transform on graphs by relying on diffusion wavelets, and provide an appropriate study of stability, which seems to highlight relevant properties of the graphs. The proposed representation seems to provide benefits compared to the previous work of Zou & Lerman, particularly regarding computational efficiency, as well as stability with respect to a metric that is perhaps more useful, though there is a dependence on the graph topology through the spectral gaps.
>
> In addition, the experiments on author attribution and source localization suggest that the resulting representation remains discriminative, in addition to providing stability to changes in graph structure.
>
> I find that these contributions provide an interesting advance in theoretical understanding of graph convolutional networks from a stability perspective, in addition to introducing a useful non-learned representation, and am thus in favor of acceptance.”
>
> RESPONSE: We thank the reviewer for the time spent in reviewing this manuscript and offering valuable insights to improve our work.
>
> *Due to the character limitation, and for a clear and detailed point-to-point answer to all the very interesting comments raised by the reviewer, we split these responses in two comments. Thank you for your understanding.*
>
> REVIEW: “Nevertheless, some parts of paper would benefit from further discussions and more clarity:
>
> - other than empirically, one aspect that's missing compared to the original study of the scattering transform is energy preservation. The authors could at least provide a discussion of whether such a property can be obtained here as well (does it depend on the spectral gap through C(beta)?)”
>
> RESPONSE: Thank you for bringing up this important point. Energy preservation in scattering representations requires a unitary wavelet decomposition. In the original scattering paper (Mallat, 2012), it was proved for a restricted class of unitary wavelets in R^d, and later extended to more general families of analytic wavelets (Wiatowski, Grohs and Bölcskei, 2017). Extending energy preservation to graph scattering thus amounts to constructing graph wavelets that are unitary and analytic. Whereas constructing unitary wavelets on graphs is easily achieved (for instance designing the wavelets in the spectral domain), the analytic property in general graphs requires a notion of frequency ‘pairings’ (analogously as one associates sine and cosine in an Euclidean domain). This is a fascinating direction of future research. We note that our diffusion wavelets are neither unitary nor analytic. However, as you correctly observed, the frame bounds do depend on the domain via the spectral gap. The lower bound is (1-beta)^2, so the smaller 1-beta is, the less “unitary” our diffusion wavelets are.
>
> We have mentioned this issue after Proposition 4.1 which establishes bounds on the energy change when applying the graph diffusion wavelet bank.
>
> REVIEW: “- what is the role of the spectral gap in the stability bounds? is this a drawback of the diffusion metric / choice of wavelets?”
>
> RESPONSE: The spectral gap is tightly linked with diffusion processes on graphs, and thus it does emerge from the choice of a diffusion metric. Graphs with values of beta closer to 1, exhibit weaker diffusion paths, and thus a small perturbation on the edges of these graphs would lead to a larger diffusion distance. The contrary holds as well. In other words, the tolerance of the graph to edge perturbations (i.e., d(G,G’) being small) depends on the spectral gap of the graph. Another interpretation of the spectral gap is as the ‘bandwidth’ of the domain. The larger the gap (thus smaller beta), the faster the domain diffusion blurs an input arbitrary signal, therefore limiting our ability to discriminate high-frequency information. A byproduct of this reduced bandwidth (small beta) are improved stability bounds.
>
> We have added a remark about this aspect at the end of section 5.

---

> > ### Author Response · Authors · 2018-11-12
> > **Response to AnonReviewer1: Part 2**
> >
> > REVIEW: “- Section 3.2 suggests that metric stability is a good way to characterize stability, by seeing deformations in Euclidian domains as a change to the ground metric. Yet, in Euclidian scattering, the same representation is applied to a deformed signal and the original signal, and stability is measured with the Euclidian metric. Can the link be made more precise, by explaining what a deformation of a signal would be on a graph, or by applying arguments from the proposed construction to the Euclidian case?”
> >
> > RESPONSE: Thank you very much for raising this very interesting point. The setting of the paper by Mallat, 2012, and Bruna and Mallat, 2013 is a continuous setting, where an image is described as a map from R^2 to R. In this sense, a deformation can be understood as a change of variables, or as a change of the underlying support. Nevertheless, the definition of deformation in that work does not intrinsically alter R^2 because it is still a continuous space. As an example, we can easily think of a deformation of a continuous image, but if we think of a digital image, we need to go through a process of resampling to model the deformed signal. This is due exclusively to the regular nature of the image grid and the existence of a sampling theorem between the continuous and the discrete spaces.
> >
> > In the case of graphs, such a resampling method is not unique, and therefore, we cannot use the same notion of continuous deformation that is used in Bruna and Mallat, 2013. We therefore opt to define a deformation in terms of the underlying discrete support of the signal under study. In the case of an image, this would be understood that deforming an image essentially alters the distance between pixels (determined by the sampling mechanism) and therefore, alters the underlying grid structure in itself. More precisely, if in the original image, all pixels are at a distance of “1” (normalized by the resolution of the image), in a deformed version of the image, we are actually altering this distance, which is analogous to altering the edges of the underlying grid. This understanding of deformation as a change in the underlying support is what motivates our definition of deformation of a graph.
> >
> > In other words, in the original Euclidean scattering, the data domain is fixed (hence the wavelet decomposition is fixed) and deformation is studied by measuring how the wavelet decomposition of x’(u) = x(u -tau(u)) varies with tau. Observe however that
> > x’ * psi(u) = int x’(v) psi(u-v) dv = int x(v - tau(v)) psi(u-v) dv = int x(v’)psi(u-v’+beta(v’)) |det (I-Dtau(beta(v’))|^(-1) dv’, which is a perturbed convolution of the same signal x on a deformed metric domain.
> >
> > We have added a comment on this intrinsic difference between deformations in the Euclidean space and in general graphs at the beginning of section 3.3 to motivate our choice of graph deformation.
> >
> > REVIEW: “- the paper is heavy on terminology from wavelets and harmonic analysis, a more detailed presentation of diffusion wavelets and related concepts such as localization would be beneficial. Also, it seems like the chosen wavelets in the construction favor spatial over frequential localization - is this due to a trade-off? if so, can it be avoided?”
> >
> > RESPONSE: We apologize for the lack of adequate coverage of signal processing related concepts such as space-frequency duality. We have reworded the presentation of diffusion wavelets in the hope of making it more clear. In general, one would require as many filter taps as the size of the graph to be able to perfectly filter each individual frequency component. Nevertheless, as with the Euclidean case, this leads to instability (very narrow frequency filters imply that a small change in the frequency content leads to a huge change in the filtering output).
> >
> > The diffusion wavelets, indeed, favor spatial localization over frequency localization. They can be constructed with just two spatial coefficients. This can indeed be avoided with more general graph wavelets, and this is the subject of our current work, where we obtained stability bounds with respect to the number of filter taps (spatial localization) used to implement the graph wavelets.
> >
> > REVIEW: “Some more detailed comments (...)”
> >
> > RESPONSE: We have carefully addressed all these comments in the revised manuscript.
> >
> > Once again, we would like to thank the reviewer for the time and effort spent in providing detailed feedback that has surely helped improve the quality of our work.

---

### Official Review · AnonReviewer3 · 2018-11-05
**Stability analysis for graph NNs**

**Rating:** 9
**Confidence:** 5

**Review:**

The paper presents in interesting and new analysis of stability of scattering transforms on graphs, when the domain (graph) is perturbed by deformations. It combines key ingredients of scattering transforms (extended here to graphs through graph diffusion wavelets), deformation of graphs (based on graph diffusion distances) and a theoretical stability analysis. Similarly to the Euclidian domain, it is shown that linear filters cannot provide representations that are simultaneously rich and stable.

Generally, the paper is pretty complete, interesting and sufficiently well presented. One might wonder if the choice of the diffusion framework for both the representation construction, and the deformation analysis, is a simplistic choice, and how similar ideas could extend to different domain deformation for example. The experiments and comparisons are also very minimal, and hard to interpret. Comparing things only with GFT and SVM is probability a 'easy' choice, with the advent of a plethora of new graph convnets architectures (GFT is probably not a 'graph baseline', as mentioned in the conclusion).

This is however an interesting work, that will likely generate exciting discussions at ICLR.

---

> ### Author Response · Authors · 2018-11-12
> **Response to AnonReviewer3**
>
> REVIEW: “The paper presents in interesting and new analysis of stability of scattering transforms on graphs, when the domain (graph) is perturbed by deformations. It combines key ingredients of scattering transforms (extended here to graphs through graph diffusion wavelets), deformation of graphs (based on graph diffusion distances) and a theoretical stability analysis. Similarly to the Euclidian domain, it is shown that linear filters cannot provide representations that are simultaneously rich and stable.”
>
> RESPONSE: Thank you very much for the time spent in reviewing this manuscript and the valuable insights provided.
>
> REVIEW: “Generally, the paper is pretty complete, interesting and sufficiently well presented. One might wonder if the choice of the diffusion framework for both the representation construction, and the deformation analysis, is a simplistic choice, and how similar ideas could extend to different domain deformation for example.”
>
> RESPONSE: We thank the reviewer for bringing up this point. We would like to let the reviewer know that the choice of diffusion distance and consequently, diffusion wavelets, is rooted in mathematical tractability. Nevertheless, the general framework for the proof is extensible to any other domain or deformation measure that offers bounds similar to those in Lemmas 5.1 and 5.2. In fact, we are currently working on extensions to general wavelet graph filters and Gromov-Hausdorff distances, as well as continuous domains.
>
> We have added comments on this after Def. 3.1 where the diffusion distance is introduced, and in the conclusions to highlight future work.
>
> REVIEW: “The experiments and comparisons are also very minimal, and hard to interpret. Comparing things only with GFT and SVM is probability a 'easy' choice, with the advent of a plethora of new graph convnets architectures (GFT is probably not a 'graph baseline', as mentioned in the conclusion).”
>
> RESPONSE: First and foremost, we would like to apologize for the statements included in the numerical section that might have accidentally misrepresented the claims made in that section. We consider this work to be of theoretical value, showing stability of deep learning constructions on graphs. In this sense, the objective of the simulation section is to illustrate, as well, the ability of the graph scattering transforms to obtain meaningful representations. By no means is our intention to claim that we perform better than the state of the art, and we apologize for any such misunderstanding that might have arised from our writing in that section.
>
> The choice of comparison methods is guided by the following rationale. We know that linear methods are unstable under deformations, and we propose an alternative representation that we prove is stable. Therefore, for the numerical sections, we chose two widely used linear models: one data-based (SVM) and one structural-based (GFT) with the aim of showing that there is no loss in the representation capability by using a stable representation. In other words, our main goal of this section is to prove that we can do as well as linear models, while guaranteeing for stability. As a byproduct, we observed that the proposed representation outperforms these linear models, but this, by no means, implies that this is the best possible solution for these problems.
>
> With respect to comparing with graph convnets, we find that the methods are conceptually different since graph scattering representations do not involve any training. In this respect, we could compare with graph convnets with a minimal training set of one (or zero) in which case scattering transforms would do better, or we could compare with graph convnets for the case in which we have a large training set, in which case graph convnets will surely fare better. Therefore, we believe these methods not to be comparable. Finally, we note that the stability analysis that we carried out for diffusion scattering also applies to Graph Convnets (see corollary 5.3). In this case, our stability bounds depend on the trainable parameters, and are generally not tight. Similarly as in the case of images, and as illustrated by adversarial examples, there is an underlying tradeoff between discriminability and stability.
>
> All in all, we have completely rewritten the simulation section making emphasis on the illustrative nature of these examples, and have carefully worded every claim to avoid any misunderstanding.
>
> REVIEW: “This is however an interesting work, that will likely generate exciting discussions at ICLR.”
>
> RESPONSE: We would like to thank the reviewer again for their time and effort spent in providing valuable feedback on our work.

---

### Official Review · AnonReviewer2 · 2018-11-20
**Interesting theoretical study with issues in motivation and experiments**

**Rating:** 6
**Confidence:** 4

**Review:**

The paper introduces scattering transforms on graphs by adopting diffusion wavelet constructions, and gives an extension of the scattering transforms to non-Euclidean domains. The main result consists of a stability analysis of the non-adaptive representation under deformation of the underlying graph metric, also defined in terms of graph diffusion.

Pros:

The study addresses the important problem of representation stability of non-Euclidean CNNs, which is a timely topic. The theoretical analysis builds an interesting connection between the diffusion graph geometry and the analysis of deep networks.

Cons:

- It is unclear what type of graph is the primary consideration, either (a) expander/small-world with large spectral gap, e.g. social network, or (b) irregular mesh embedded in a Euclidean space or on an intrinsic manifold - as originally considered in diffusion wavelet, and in the Euclidean CNN/scattering transform theory - which often fails to present a spectral gap. The formula suggests (b) while the analysis and experiments point to (a). A coverage of both cases is unlikely.

- The deformation considered is proposed in terms of the graph metric perturbation, which appears to lack sufficient motivation. This is not apparent from the experimental results (How is the x-axis of plots in Figure 1 related to "perturbation"?). Will such deformation reduce to that as being considered for irregular meshes in computer vision applications e.g. in [1]? The proposed model would be more convincing if the class of "covered deformations" can be clarified and the relevance to practices of non-Euclidean CNNs can be better addressed.

- The experimental section lacks performance and comparison in a controlled environment, e.g., on synthetic data with more samples to show statistical significance. There also appears to be a gap between theory and experiment: can the dependence on spectral gap be empirically supported, even only qualitatively, e.g., by a comparison on small world graph v.s. others?

Overall, the recommendation is still in favor of acceptance. Finally, the reviewer would like to raise the following questions:

- As the result is presented as an extension of the Euclidean scattering transform, will the stability result recover the traditional one in Mallat et al. when the underlying graph is a regular grid (though the definition of deformation differs in appearance)?

- What about the computational efficiency, scalability and storage cost of the algorithm? It would also clarify the computational procedure by providing an algorithm box or release of code. However, assuming that the focus of the paper is on theory, computation is a relatively minor point.

[1] Boscaini, D., Masci, J., Rodolà, E., & Bronstein, M. (2016). Learning shape correspondence with anisotropic convolutional neural networks. In Advances in Neural Information Processing Systems (pp. 3189-3197).

---

> ### Author Response · Authors · 2018-11-26
> **Response to AnonReviewer2: Part 1**
>
> REVIEW: “The paper introduces scattering transforms on graphs by adopting diffusion wavelet constructions, and gives an extension of the scattering transforms to non-Euclidean domains. The main result consists of a stability analysis of the non-adaptive representation under deformation of the underlying graph metric, also defined in terms of graph diffusion.”
>
> RESPONSE: We would like to thank the reviewer for taking the time to review this manuscript and providing valuable feedback.
>
> REVIEW: “Pros:
>
> The study addresses the important problem of representation stability of non-Euclidean CNNs, which is a timely topic. The theoretical analysis builds an interesting connection between the diffusion graph geometry and the analysis of deep networks.
>
> Cons:
>
> - It is unclear what type of graph is the primary consideration, either (a) expander/small-world with large spectral gap, e.g. social network, or (b) irregular mesh embedded in a Euclidean space or on an intrinsic manifold - as originally considered in diffusion wavelet, and in the Euclidean CNN/scattering transform theory - which often fails to present a spectral gap. The formula suggests (b) while the analysis and experiments point to (a). A coverage of both cases is unlikely.”
>
> RESPONSE: We thank the reviewer for bringing up this very fundamental point. We acknowledge that the provided stability results depend on the spectral gap which makes its application dependent on the particular graph. As the reviewer points out, there are graphs that exhibit a large spectral gap while others exhibit a small spectral gap, thus making our stability bounds less effective.
>
> We would like to remark that the only connected graph to present no spectral gap is the one that contains a non-trivial bipartite component (Chung, 1997). But since we are considering lazy diffusion (i.e., we use as a generator T = (I +A)/2)  these degenerate cases are avoided. In particular, for regular graphs with self-loops, it is known that the spectral gap is lower bounded by (dn)^(-2), where d is the degree. That said, it is true that this gap is not uniform in the size of the graph, and the rate n^(-2) reflects the fact that as the domain grows, so does the number of effective scales. We emphasize that this situation reflects a limitation of our current bound, and that it is likely that one can improve it (in particular, in eq. 23) by leveraging regular degree structures (as in grids). We will work on this direction and keep the reviewer updated.
>
> The distinction between these two extremal regimes (expanders vs. manifolds) is also fundamental when it comes to our domain deformation model. Whereas in the former the natural notion of deformation is necessarily intrinsic (and we chose to use diffusion distances but other intrinsic notions are possible), in the latter the deformations can be ‘inherited’ from the extrinsic space, and typically one can get finer analysis using Euclidean deformations. For instance, in our framework we have not defined the ‘smoothness’ of a deformation, which plays a crucial role in Euclidean Scattering to avoid a deformation bound that degrades with the bandwidth of the signal. This is an important direction for future work.
>
> We have added relevant comments after eq. (3) and before Proposition 4.1.

---

> > ### Author Response · Authors · 2018-11-26
> > **Response to AnonReviewer2: Part 2**
> >
> > REVIEW: “- The deformation considered is proposed in terms of the graph metric perturbation, which appears to lack sufficient motivation. This is not apparent from the experimental results (How is the x-axis of plots in Figure 1 related to "perturbation"?). Will such deformation reduce to that as being considered for irregular meshes in computer vision applications e.g. in [1]? The proposed model would be more convincing if the class of "covered deformations" can be clarified and the relevance to practices of non-Euclidean CNNs can be better addressed.”
> >
> > RESPONSE: Thank you for this important remark. The motivation behind the choice of graph metric perturbation is to define a notion that is intrinsic, i.e. not relying on any extrinsic Euclidean embedding of the domain. That said, the rationale behind the choice of diffusion metrics is primordially of mathematical nature (tractability). With respect to the simulations, we intended to present graph perturbations that would be rooted in practical situations (edge failures -unfriending- in the case of the social network graph, and faulty graph modeling -limited available data to build the supporting graph- in the case of authorship attribution) that still illustrate the stability property of the proposed scattering transform. These perturbations are related to diffusion metrics in the sense that the possible diffusion paths are being altered between the different underlying graphs that support the data.
> >
> > With regards to the relationship between our diffusion-based deformation model and those in computer vision/graphics, we emphasize two points. (i) Our notion is purely intrinsic, so it captures the isometric invariances enjoyed by intrinsic shape/manifold representations (including some non-euclidean CNN constructions). (ii) Our notion does not leverage the Euclidean ambient space to define smoothness of a deformation field, which could be used in irregular meshes/manifolds to control the impact of the deformation on scattering coefficients specifically at each scale and thus improve the stability bound.
> >
> > REVIEW: “- The experimental section lacks performance and comparison in a controlled environment, e.g., on synthetic data with more samples to show statistical significance. There also appears to be a gap between theory and experiment: can the dependence on spectral gap be empirically supported, even only qualitatively, e.g., by a comparison on small world graph v.s. others?”
> >
> > RESPONSE: We would like to thank the reviewer for the suggestion. We have added to the simulation section a first experiment where we create synthetic small world graphs and compute the scattering representation of random signals supported in these graphs. More specifically, we generate graphs with the same parameters p_SW (edge probability) and q_SW (rewiring probability) in order to fix a spectral gap (which is related to p_SW), and then compute the difference between the scattering representation of the same signal over these graphs. Then, we change p_SW to change the spectral gap. We observe that, indeed, as the spectral gap get smaller (beta gets closer to 1), the representation difference grows larger, showing that graphs with smaller spectral gap give place to less stable representations. Please, see this section for more details.

---

> > > ### Author Response · Authors · 2018-11-26
> > > **Response to AnonReviewer2: Part 3**
> > >
> > > REVIEW: “Overall, the recommendation is still in favor of acceptance. Finally, the reviewer would like to raise the following questions:
> > >
> > > - As the result is presented as an extension of the Euclidean scattering transform, will the stability result recover the traditional one in Mallat et al. when the underlying graph is a regular grid (though the definition of deformation differs in appearance)?”
> > >
> > > RESPONSE: Thank you very much for bringing up this very interesting aspect. The simple answer is no, because we do not have a notion of smoothness in the deformation field currently. In our case, we only see the deformation through its impact on the metric of the domain. In Euclidean scattering, on the other hand, one can relate the wavelet acting on signals at a specific scale with the regularity of the deformation field at that scale, and use a sophisticated result from harmonic analysis (the Cotlar-Stein quasi-orthogonality lemma) to bound the interactions across different scales. It is unclear to the authors that such result could be extended to non-Euclidean domains, and is currently part of our ongoing work on this subject.
> > >
> > > REVIEW: “- What about the computational efficiency, scalability and storage cost of the algorithm? It would also clarify the computational procedure by providing an algorithm box or release of code. However, assuming that the focus of the paper is on theory, computation is a relatively minor point.”
> > >
> > > RESPONSE: To compute any given coefficient, we have to compute U(rho Psi)^{k} x. Applying Psi is just weighing two neighborhoods, which ignoring any kind of sparsity, entails a computation cost of JN^2. Applying U entails JN operations. The total cost is thus m(JN^2+JN) where m is the depth of the representation.
> > >
> > > REVIEW: “[1] Boscaini, D., Masci, J., Rodolà, E., & Bronstein, M. (2016). Learning shape correspondence with anisotropic convolutional neural networks. In Advances in Neural Information Processing Systems (pp. 3189-3197).”

---

### Author Response · Authors · 2018-11-26
**Updated revision: Summary of changes**

We have uploaded the revised version of the manuscript.

Major changes include:

- New discussion on the role of the spectral gap on the obtained bounds;
- New discussion on the choice of diffusion metrics to measure support deformations;
- Rewriting of the experimental section to highlight its objective of illustrating discriminative power;
- New experiments testing the representation difference sensitivity to the spectral gap;
- New discussion on the manifold/expander type of graphs;
- Fixing of several typos in the proofs;
- Address of all the comments raised by the reviewers.

We would like to thank, once more, to all reviewers for taking the time to provide valuable feedback that has certainly improved the quality of the manuscript.

---

### Meta-Review · Area_Chair1 · 2018-12-15
**Unanimously accept for ICLR publication.**

**Confidence:** 5
**Recommendation:** Accept (Poster)

**Metareview:**

The paper gives an extension of scattering transform to non-Euclidean domains by introducing scattering transforms on graphs using diffusion wavelet representations, and presents a stability analysis of such a representation under deformation of the underlying graph metric defined in terms of graph diffusion.

Concerns of the reviewers are primarily with what type of graphs is the primary consideration (small world social networks or point cloud submanifold samples) and experimental studies. Technical development like deformation in the proposed graph metric is motivated by sub-manifold scenarios in computer vision, and whether the development is well suitable to social networks in experiments still needs further investigations.

The authors make satisfied answers to the reviewers’ questions. The reviewers unanimously accept the paper for ICLR publication.